# Functional In Vivo Screening Identifies microRNAs Regulating Metastatic Dissemination of Prostate Cancer Cells to Bone Marrow

**DOI:** 10.3390/cancers15153892

**Published:** 2023-07-31

**Authors:** Tina Catela Ivkovic, Helena Cornella, Gjendine Voss, Anson Ku, Margareta Persson, Robert Rigo, Sofia K. Gruvberger-Saal, Lao H. Saal, Yvonne Ceder

**Affiliations:** 1Department of Laboratory Medicine, Division of Translational Cancer Research, Lund University, 223 81 Lund, Sweden; tina.catela_ivkovic@med.lu.se (T.C.I.); gvoss@mbl.edu (G.V.); margareta.persson@med.lu.se (M.P.); 2Division of Molecular Medicine, Ruder Boskovic Institute, 10000 Zagreb, Croatia; 3Department of Translational Medicine, Lund University, 205 02 Malmö, Sweden; anson.ku@nih.gov; 4Division of Oncology and Pathology, Lund University, 223 81 Lund, Sweden; robert.rigo@med.lu.se (R.R.); sofia.gruvbergersaal@skane.se (S.K.G.-S.); lao.saal@med.lu.se (L.H.S.)

**Keywords:** prostate cancer, microRNAs, metastases, bone marrow, therapeutics

## Abstract

**Simple Summary:**

Prostate cancer confined to the prostate is not a cause of death, but when the tumour has metastasized, it is lethal. By using a functional screening method, we identified a microRNA that increased the metastatic spread to bone. We then found that the levels of this microRNA were changed in patients with bone metastases. Our study suggests that this is due to effects on cell growth and colony-formation capacity in the bone microenvironment.

**Abstract:**

Distant metastasis is the major cause of cancer-related deaths in men with prostate cancer (PCa). An in vivo functional screen was used to identify microRNAs (miRNAs) regulating metastatic dissemination of PCa cells. PC3 cells transduced with pooled miRZiP™ lentivirus library (anti-miRNAs) were injected intraprostatic to 13 NSG mice followed by targeted barcode/anti-miR sequencing. PCa cells in the primary tumours showed a homogenous pattern of anti-miRNAs, but different anti-miRNAs were enriched in liver, lung, and bone marrow, with anti-miR-379 highly enriched in the latter. The bone metastasis-promoting phenotype induced by decreased miR-379 levels was also confirmed in a less metastatic PCa cell line, 22Rv1, where all mice injected intracardially with anti-miR-379-22Rv1 cells developed bone metastases. The levels of miR-379 were found to be lower in bone metastases compared to primary tumours and non-cancerous prostatic tissue in a patient cohort. In vitro functional studies suggested that the mechanism of action was that reduced levels of miR-379 gave an increased colony formation capacity in conditions mimicking the bone microenvironment. In conclusion, our data suggest that specific miRNAs affect the establishment of primary tumours and metastatic dissemination, with a loss of miR-379 promoting metastases in bone.

## 1. Introduction

Prostate cancer (PCa) is estimated to be the most prevalent cancer type in men worldwide. While confined to the prostate gland, the disease can be successfully managed with established treatment options [1], giving a five year survival rate of almost 100%. Once the PCa has spread, the five-year survival rate drops to 32% [2]. Bone is the most common metastatic site (~80%), and at present, there are no curative treatments available [3,4]. 

Metastatic dissemination is an inefficient, complex, and dynamic process. A key feature enabling successful metastatic dissemination is cellular plasticity. Cellular plasticity is believed to be directed by reciprocal feedback loops between microRNAs (miRNAs) and, e.g., miR-200 and the ZEB family [5], and the miR-34 family and, e.g., SNAIL1 [6]. It is conceivable that analogous feedback loops between miRNA and their EMT/stemness-inducing targets regulate tumour cell plasticity in response to exposure to novel microenvironments. Due to the weak complementarity requirement for miRNA binding, each miRNA can recognize many mRNA targets, allowing versatile and dynamic regulation that can vary in different cellular settings [7]. Changes in miRNA levels in primary prostate tumours, circulation and metastases have been identified; thus, miRNAs are interesting potential prognostic markers as well as therapeutic targets. Our team has previously found the expression of individual miRNAs in the primary prostate tumour to correlate with metastatic disease and to be an independent predictor of metastatic events occurring 0.5–10 years after the removal of the primary tumour [8]. We have also shown that, e.g., miR-34c and miR-205 reduce motility and invasive capabilities of tumour cells, and show an inverse correlation to occurrence of metastases in patients [9,10].

However, the level of miRNA deregulation does not directly imply therapeutic potential; deregulation can result from the metastatic processes rather than driving the progression, and the level of deregulation does not necessarily reflect the impact of an individual molecule; a small change in the levels of a key molecule can be more influential than a larger change in levels of a more peripheral molecule. In order to identify novel therapeutic targets, we employed an in vivo functional screening approach to identify individual miRNAs potentially inhibiting the metastatic process. Primarily, we performed a loss-of-function screening, where anti-miRNAs decreasing the levels of miRNA with inhibiting potential can be identified, as this results in increased metastatic dissemination. This was followed by in vitro and in vivo verifications and validation in a clinical cohort.

## 2. Materials and Methods

### 2.1. Patient Samples 

Patient samples were collected at Umeå University hospital. The cohort has previously been described [11,12]. Briefly, it comprises 55 freshly frozen bone metastasis samples from PCa patients undergoing surgery for spinal cord compression, 12 prostate tumour samples from radical prostatectomies, and 13 samples of non-malignant adjacent prostate tissues. The project was carried out in accordance with the Helsinki Declaration. It was approved by the Local Ethics Committee, and all participants gave their informed consent.

### 2.2. Cell Lines

PCa cell lines PC3 (CRL-1435™) and 22Rv1 (ECACC 05092802) were obtained from American Type Culture Collection and European Collection of Authenticated Cell Cultures, respectively. The cells were maintained according to the supplier’s recommendations unless otherwise specified. The cell lines were regularly tested for mycoplasma contamination and authenticated, most recently in February 2023 by Eurofin Genomics. The human primary mesenchymal stem cells (a kind gift from Dr. Stefan Scheding) were expanded in StemMACS expansion medium (Miltenyi, Bergisch-Gladbach, Germany) and differentiated for three weeks to osteoblasts in low-glucose DMEM with 10% FBS, 10 mM β-glycerophosphate, 0.05 mM IPL-ascorbic acid, and 0.1 µM dexamethasone (Sigma-Aldrich, Steinheim, Germany). The differentiation to osteoblasts was confirmed by 10 mg/mL Alizarin Red S staining (Sigma Aldrich, Saint Louise, MO, USA). After at least 48 h, the osteoblast-conditioned medium (OB-CM) was collected and cleared by centrifugation at 900× *g*. The experiments were carried out in 50% conditioned and 50% fresh medium.

### 2.3. RNA Extraction, Reverse Transcription and qRT-PCR 

Small RNAs were extracted from the patient samples using the Allprep protocol, as described in [12] followed by further enrichment and purification of short RNA specimens using the RNeasy MinElute Cleanup kit (Qiagen, Hilden, Germany). Total RNA from cell lines was extracted using Trizol (Ambion, Thermofisher Scientific, Waltham, MA, USA) according to the manufacturer’s protocol. The samples were quantified on a Nanodrop 2000 Spectrophotometer (Thermo Fisher Scientific). Four nanograms of RNA from each sample were used for conversion to cDNA. Reverse transcription was performed using the TaqMan™ Advanced miRNA cDNA Synthesis Kit (Applied Biosystems, Foster City, CA, USA), and expression levels of miR-379-5p were analysed using quantitative real-time PCR with TaqMan™ Advanced miRNA assay (478077_mir) according to the manufacturer’s protocol. All reactions were run in triplicate and the averages of the three Ct values were used for further calculation. Undetermined Ct values were set to 40. The results were normalized to the average expression values of hsa-miR-331-3p (478323_mir), hsa-miR-186-5p (477940_mir), and hsa-miR-361-5p (478056_mir) (Applied Biosystems). The comparative Ct method (2^−ΔCt^) was used for analysis [13]. 

### 2.4. Transduction of Cells with miRZiP Library 

Cell lines were transduced using the anti-miRNA library, miRZip™ Lentivirus Pool of Anti-MicroRNAs (MZIPPLVA-1, System Biosciences, Palo Alto, CA, USA), following the manufacturer’s protocol. This anti-miRNA library is a pool of 167 anti-miRNA shRNA clones. The vector contains a copGFP (green fluorescent protein) under the CMV promoter allowing monitoring of tumour progression, while the integrated anti-miRNA itself acts as a simplified barcode. Individual inserts within the pool are designed to knockdown a miRNA each, or a scrambled control sequence (SC) without a target. To reduce the risk of several inserts per cell the multiplicity of infection (MOI) was optimized for each cell line, aiming at a transduction efficiency of 20% (MOI = 1.3 for PC3 and 0.1 for 22Rv1). Transduction efficiency was monitored by GFP signal, and the transduced cells were selected using 0.5 μg/mL (22Rv1) and 1 μg/mL puromycin (Invivogen, Toulouse, France) for 22Rv1 and PC3, respectively. To exclude the dominance of specific inserts from the pool, the library-containing cell lines (LB-22Rv1 and LB-PC3) were sequenced using targeted next generation sequencing with lentivector-specific primers (Appendix A).

### 2.5. In Vivo PCa Model of Metastatic Migration

NOD scid gamma (NSG) mice (kind gift from Dr. Daniel Bexell) were used for all animal models. For the functional anti-miRNA library screen, 1 × 10^6^ LB-PC3 cells resuspended in 20 µL of medium and colour dye were injected into the dorsolateral lobe of the prostate of 14 seven-week-old mice using a 30-gauge needle. Mice were checked daily and weighed twice a week. Five weeks after injection, 13 mice were euthanized after losing 10% of their weight, and primary tumours as well as potential metastatic sites—livers, lungs, and bone marrow from femurs—were collected. One mouse was found dead at the same time; from this, only the primary tumour was retrieved. The presence of metastases was confirmed using GFP fluorescence. All samples were stored in RNA^TM^ Later (Thermofisher) according to manufacturer’s recommendations until further use. 

The protocol for the intracardiac model was adapted from Campbell et al. and Prensner et al. [14,15]. Seven weeks old mice were anaesthetized with isoflurane and maintained under anaesthesia during the procedure. Then, 5 × 10^5^ 22Rv1-anti-miR-379 or 22Rv1-SC cells in a total volume of 100 µL were injected into the left cardiac ventricle of each mouse. Mice were checked up daily and weighed twice a week. Mice were euthanized upon weight loss after approximately five weeks, and potential metastatic sites—livers, lungs, and bones (left humerus and left femur)—were collected. Samples were formalin-fixed and paraffin-embedded. The bones were decalcified in 10% EDTA for 96 h prior to embedding in paraffin. All mouse experiments were approved by the Local Ethical Committee. 

### 2.6. DNA Extraction and PCR 

DNA from cell lines was extracted and purified using QIAmp DNA kit (Qiagen). The mouse tissue samples collected from the orthotopic library screen model were homogenized and used for DNA extraction. DNA was extracted and purified using AllPrep^®^ DNA/RNA/miRNA Universal Kit (Qiagen). DNA was quantified using Nanodrop 2000 (Spectrophotometer (Thermo Fisher Scientific)). Two hundred ng of DNA was amplified in a nested PCR, with a pair of custom-created EXT primers (Appendix A) and miRZip primers in the second step, using Taq DNA polymerase (Applied Biosystems, Foster City, CA, USA). The product was purified using QIAquick PCR Purification Kit (Qiagen). Purified PCR products were quantified using Nanodrop 2000 and Qubit dsDNA HS Assay Kit on Qubit Fluorometer (all Thermo Fisher Scientific).

### 2.7. DNA Sequencing 

Targeted DNA sequencing was performed with 15 ng of purified anti-miRNA library-specific product from a nested PCR reaction with miRZiP primers. The library was prepared using Illumina^®^ TruSeq^®^ ChIP Sample Preparation Kit (Illumina, San Diego, CA, USA) according to the manufacturer’s protocol. Instead of the gel purification step, we conducted size-selection using AMPure XP Beads (Beckman Coulter, Brea, CA, USA) in ratio beads:sample of 0.8:1. Final libraries were analysed on a Qubit Fluorometer using a Qubit dsDNA HS Assay Kit (Life Technologies, Thermo Fisher Scientific, Carlsbad, CA, USA) and on a Bioanalyzer 2100 instrument using a High Sensitivity DNA chip (Agilent Technologies, Santa Clara, CA, USA). Samples were adjusted to 2 nM and pooled for sequencing. DNA sequencing was done on MiSeq instrument according to manufacturer’s instructions (Illumina, San Diego, CA, USA).

### 2.8. Immunohistochemical Analysis 

Formalin-fixed paraffin-embedded tissue sections were stained for GFP using primary specific antibody anti-TurboGFP (Invitrogen, Thermo Fisher Scientific, Carlsbad, CA, USA; 1/100 dilution). In brief, sections were first deparaffinized and rehydrated. Antigen retrieval was carried out using 10 mM Citrate buffer pH 6.0 (Dako, Agilent Technologies, Santa Clara, CA, USA). Primary antibody was incubated overnight at 4 °C. Specific antibody binding of anti-TurboGFP was detected using DAB Vector peroxidase substrate Kit (Vector Laboratories, Inc., Burlingame, CA, USA) and counterstained with Hematoxylin (Mayers HTX PLUS Histolab products AB, Askim, Sweden). Each slide was examined using an Olympus Dp80 BX63 Fluorescent microscope (LRI, Lund, Sweden) in the entire section area for the presence of GFP in library-containing cells. 

### 2.9. RNA Immunoprecipitation Sequencing (RIP-Seq)

The protocol has been previously described [16]. In brief, cells were harvested 48 h after transfection with miR-379-5p mimics or negative control, and the antibody against human AGO2 (Sigma Aldrich, Saint Louise, MO, USA) was used. The samples underwent RNA extraction for downstream sequencing analysis. 

### 2.10. RNA Sequencing 

RNA sequencing was performed at the Center for Translational Genomics, Lund University and Clinical Genomics Lund, SciLifeLab, according to a protocol adapted from Saal et al. [17]. In brief, 100 pg of total immunoprecipitated RNA was fragmented to ~240 bp by Ambion buffered zinc fragmentation reagents (Ambion Thermo Fisher Scientific, Waltham, MA, USA). Purified dscDNA was further processed in the same way as for targeted DNA sequencing, using the Illumina^®^ TruSeq^®^ ChIP Sample Preparation Kit (Illumina, San Diego, CA, USA).

### 2.11. In Vitro Functional Analyses

The soft agar colony formation/anchorage-independent growth assay was performed as previously described [18]. For construction of stable anti-miRNA-expressing cells, individual colonies from the LB-PC3/LB-22Rv1 libraries were picked from colony formation assays or by seeding one cell per well in a 96-well plate. In both approaches, the clones were expanded and then verified with Sanger sequencing LightRun, using the GATC method (Eurofins, Jena, Germany) with miRZip primers. For downstream experiments, three individual clones containing the same insert were combined in the cell number ratio 1:1:1. For transient cell transfection we used 120 nM miRIDIAN microRNA Mimic (Dharmacon, Lafayette, CO, USA) and Oligofectamine reagent (Invitrogen, Thermo Fisher Scientific) according to the manufacturer’s instructions. For normalization purposes we also transfected cells with miRIDIAN microRNA Mimic Negative Control (Dharmacon) under the same conditions. The sulforhodamine B (SRB) colorimetric assay was performed as previously described [19]. Readout of cells cultured overnight was used for normalisation to compensate for the number of seeded cells. The migration experiment was conducted as previously described [9] but the 22Rv1-anti-miR-379 and 22Rv1-SC clones were plated into Transwell cell culture inserts 8.0 uM (Merck, Kenilworth, NJ, USA) at the density of 1 × 10^5^ cells/cm^2^ and incubated for 18 h. The intensity was analysed using a microplate reader Synergy 2 BioTek Instrument (Agilent, Santa Clara, CA, USA). 

### 2.12. Data Processing and Statistical Analysis 

For the analyses of the miRZiP library sequencing data, the number of read pairs containing each anti-miRNA was counted and divided by the total number of matched reads for that sample to obtain a percentage of each specific anti-miRNA sequence. Low abundance tags, where the number of counts across all samples was below 1%, were filtered, and the samples and anti-miRNAs were clustered using Ward’s minimum variance method using the hclust function in R 3.4.0 and plotted as heatmaps. RNA expression was analysed as log-transformed data. For continuous data obtained from patient samples, a normality test was applied [20] and the appropriate statistical test, *t*-test or Kruskal–Wallis trend test, was used for further analysis. For the in silico analyses of the TCGA paired dataset, the Wilcoxon matched-pairs signed-rank test was used with the Sidak–Bonferroni correction to adjust for multiple comparisons. All in vitro experiments were analysed using two-tailed unpaired Student’s *t*-tests. Gene ontology analysis was performed using DAVID (http://david.abcc.ncifcrf.gov, accessed on June 2020). For all statistical tests, α = 0.05 was chosen as the significance level. Analyses were carried out using GraphPad Prism software 9 [21].

## 3. Results

### 3.1. Transduction of PCa Cells with Anti-miR Library

The androgen-independent PCa cell lines PC3 and 22Rv1 were transduced with the pooled miRZiP™ lentivirus library targeting 178 miRNAs. Stable incorporation of anti-miRs constructs to the cell lines, hereafter denoted LB-PC3 and LB-22Rv1, were confirmed using targeted DNA sequencing. LB-22Rv1 was found to contain 113 different anti-miRs and to be dominated by anti-miR-135b (44%), while LB-PC3 contained 123 different anti-miRs and showed a more heterogeneous pattern, with the highest frequency of 10% for anti-miR-144 (Figure 1B). Based on this the LB-PC3 was selected for the in vivo functional screen. 

### 3.2. In Vivo Functional Library Screening

The integration of the anti-miR sequences in the cellular DNA allows identification of anti-miRNAs promoting tumour formation and metastatic spread in vivo, as illustrated in Figure 1A. One million LB-PC3 cells were injected into the dorsolateral lobe of the prostate of 14 male NSG mice. This number allows representation of each anti-miRNA and minimises insertion site-specific effects. 

Primary prostate tumours, and the most common metastatic sites—livers, lungs, and bone marrow—were collected. Microscopic detection of GFP-expressing cells showed tumour take and presence of metastases in liver, lung, and bone marrow in all mice. The anti-miRNAs presence was then evaluated with next-generation sequencing. The sequencing results revealed that the primary tumours contained on average 97 (73–111) different anti-miRNA LB-PC3 cells, compared to the 123 different anti-miRNA LB-PC3s in the injected cell pool, and notably different anti-miRNAs were enriched in the primary tumours (Figure 1B). The sequence analyses revealed a strikingly consistent pattern of an anti-miRNA subset driving the establishment of the primary xenograft tumours; with 21 being enriched (fc > 2) and 60 depleted (fc < 0.5), as shown in the heat map in Figure 1C and Appendix A. The most frequent anti-miRNA in the primary tumours, anti-miR-493, was present in 13% of the cells, corresponding to a 10-fold enrichment compared to its presence in the injected cell pool (Figure 1B and Appendix A). The highest enriched anti-miRNA compared to injected cell pool was miR-95 which was enriched 150-fold but present in only 0.2% of the primary tumour cells. This suggests that the presence of individual anti-miRNAs provides an advantage for primary tumour establishment in vivo, which is not just a reflection of the effect on cellular proliferation rate in vitro. Exploring the paired TCGA dataset constituting 52 patients, 58 of the 178 miRNAs targeted in the miRZiP™ library were expressed at significantly higher levels in PCa primary tumours than in adjacent normal prostate tissue, and 10 at lower levels (Appendix A). The only miRNA enriched in the primary tumour cell in our screen and found to be significantly lower in the primary prostate tumour in the paired cohort was miR-379.

### 3.3. Metastatic Dissemination

We investigated intra-mouse metastatic heterogeneity and found that micro-dissecting four different liver metastases from one mouse gave four different anti-miRNA profiles (Appendix A). Each metastasis contained more than one anti-miR. The non-dominant anti-miRs were not identical to the other anti-miRs present in close-by foci, contraindicating contamination. Our deduction was that the four metastases in this liver were the result of independent events. We therefore chose to homogenize the organs to obtain the average frequency of dissemination of the different anti-miRNA LB-PC3 cells to each organ instead of microdissection of each metastasis. Of note is that the anti-miR dominating the primary tumour MZIP195 does not show up in any of the metastases. 

The anti-miRNA distribution in the metastases was found to be different from that found in the primary tumours, but the patterns within the specific organs were homogenous across mice (Figure 2A). In the clustering analyses, the samples of the same organ from different mice clustered closer together than different organ samples originating from the same mouse, both including and excluding the primary tumours (Figure 2A and Appendix A). The scramble-containing cells act as a control for the inherent metastatic dissemination, that is, as a background control. At all three metastatic sites, anti-miRNAs were detected at a higher average frequency than the scramble. This implies that individual miRNAs impact the homing and/or establishment of PCa cells to different metastatic sites. The most frequently found anti-miRNA found in the livers was anti-miR-135b detected in 44.4% of LB-PC3 cells (14-fold enriched compared to the corresponding primary tumours), in the lungs anti-miR-23b was found in 17.4% (four-fold enrichment) and in the bone marrows anti-miR-379 was found in 28.6% (over 100-fold enrichment) (Figure 2B–G). 

### 3.4. In Vitro Functional Analyses

As bone is the most common metastatic site for PCa, we focused our further studies on this specific environment. To elucidate why miR-379 seems to preferentially affect the metastatic dissemination to bone, we performed functional studies in vitro, where human primary bone marrow mesenchymal stem cells differentiated into osteoblasts (OB) were used to partly mimic the bone environment. In vitro 22Rv1-anti-miR-379 showed significantly higher levels of cell growth in both normal growth medium and OB-CM as determined by SRB assays (Figure 3A). Another event that could affect preferential metastatic dissemination to bone is migration; however, we observed no effect of anti-miR-379 on cell migration towards osteoblasts (Appendix A). We then studied the ability to expand and form a colony from a single cell, something that is essential for establishment at novel sites; 22Rv1-anti-miR-379 in OB-CM formed significantly more colonies both in comparison to 22Rv1-SC and to 22Rv1-anti-miR-379 grown in normal medium (Figure 3B and Appendix A). To further study this key function, colony formation assays were performed with anti-miR library-transduced LB-22Rv1 cells in normal growth medium and OB-CM. The LB-22Rv1 cells grown in normal growth medium were highly enriched with the anti-miR-135b-containing cells, representing over 70% of clones in the colony formation assay (Figure 3C). Notably anti-miR-379-containing cells formed no colonies in this setting. However, when performing colony formation assays in OB-CM, the miR-135b dominance was reduced and instead a striking enrichment of colonies containing anti-miR-379 and anti-miR-15b was seen (Figure 3D). These results are strikingly close to what was seen in the mouse library screen in bone metastases, suggesting that colony formation is a key function for the metastatic dissemination in the model system we use.

Anti-miR-379 was both the most frequent and most enriched anti-miRNA in the bone metastases in the in vivo screen using the PC3 cell line, as well as the most enriched anti-miRNA in the in vitro colony formation assay using 22Rv1 in OB-CM, but not in the normal medium. This prompted us to further study the molecular mechanisms of action of miR-379 in the two different settings. We performed Ago2-immunoprecipitation in 22Rv1-anti-miR-379 and 22Rv1-SC grown in normal, and OB-CM followed by RNA-seq. Between 2 and 6 × 10^7^ unique reads were detected in the analysed samples, and the main constituent was, as expected, protein-coding mRNA. Targets of miR-379 are expected to be associated less with Ago2 in anti-miR-379 cells compared to sc cells. The number of Ago2-bound mRNAs detected at significantly lower levels in 22Rv1-anti-miR-379 in comparison to 22Rv1-SC was 465 in normal medium and 263 in OB-CM (Appendix A). Interestingly, only 93 transcripts were targeted by miR-379 in both growth conditions (Figure 3E). Gene ontology analyses on the differentially expressed genes, that were decreased in the cells containing anti-miR-379 in the different settings, showed regulation of transcription and proliferation to be enriched in the normal growth medium, while cell division, Notch signalling, and hormone response were enriched in the OB-CM (Figure 3F). The 93 differentially expressed genes that were present in both growth conditions were enriched for signal transduction, positive regulation of transcription from RNA polymerase II promoter and negative regulation of cell proliferation. 

### 3.5. Metastatic Dissemination to Bone

To further investigate whether reduced levels of miR-379 can increase metastatic potential, we utilised a cell line, 22Rv1, with lower inherent metastatic potential than PC3 cells. We injected mice intracardially with 22Rv1-anti-miR-379 or 22Rv1-SC. The five mice injected with 22Rv1-anti-miR-379 cells all had bone metastases in both the femur and humerus, while none of the five mice injected with 22RV1-SC cells did, as detected with GFP fluorescence and immunohistochemistry (Figure 4A,B).

The findings were further substantiated in a clinical cohort of PCa patients collected at Umeå University Hospital, consisting of bone metastases obtained from 55 patients, primary tumour samples from 12 separate patients, and 13 samples of non-malignant adjacent prostatic tissue. The analysis showed miR-379 expression to be significantly lower in the bone metastases (Figure 4C). 

## 4. Discussion

The success rate of individual tumour cells in forming clinical metastases is very low. Interestingly, the data we show here indicates that the miRNA content contributes to give certain cells advantages for successful development of clinical metastases. If the anti-miRNAs did not affect the survival and proliferation rate of the cells, the frequencies would not change over time and would stay at around 0.8%, but the large difference in anti-miRNA frequencies both in the primary tumours and different metastatic locations indicates that the presence of individual anti-miRNAs gave a growth advantage or disadvantage. Understanding these differences and background mechanisms is very important for improvement of PCa patient prognosis but can also open possibilities for development of novel treatment strategies or to refine the already existing ones. The advantage of using a miRZiP™ library that is integrated in cellular DNA is that the anti-miR can act as a barcode, allowing tracking in vivo. The level of blocking of the anti-miRs cannot be evaluated directly as the binding to its target can be lost, for example, upon heating in a PCR reaction; however, the degree of blocking can be assumed to be less than if a CRISPR-CAS6 approach would have been used. This could result in less dramatic biological effects, but also mimics more the natural events where miRNAs are often decreased but rarely deleted. 

Analysing the primary tumours of the mice injected with the LB-PC3 miRZiP™ transduced cells, showed a multitude of anti-miRNAs suggesting a multicellular origin of the tumours. Even though each primary tumour comprises cells containing different anti-miRNAs, the pattern of anti-miRNAs in the primary tumours is strikingly homogeneous across different mice, suggesting a biological effect of individual anti-miRNAs on tumour formation. The top four anti-miRNAs detected in primary tumours were anti-miR-493, anti-miR-195, anti-miR-23b, and anti-miR-135b; notably all of the corresponding miRNAs have previously been reported to be downregulated in PCa and to act as tumour suppressors, supporting the validity of our model system [22,23,24,25]. Further, our results from the microdissection of different liver macro-metastases gave insight into the heterogeneity of the metastases. The results indicate that the metastases can be of both unicellular and multicellular origin in this model system. We then choose to look at all tumour cells in the distant organs, the strength of this strategy is that we can pick up all anti-miRs contributing to metastases, the weakness is that we also pick up dormant disseminated tumour cells. In liver metastases, miR-135b was further enriched compared to primary tumours and detected in over 40% of the prostate cells. In vitro, we observed a growth advantage to LB-PC3 cells with anti-miR-135b, but an even greater advantage to the LB-22Rv1 where cells with anti-miR-135b constituted 44% after ten passages, and at a later stage represented 56%. In the colony formation assay for LB-22Rv1 in normal medium, the anti-miR-135b cells ended up constituting 70% of all colonies. The stronger effect in the AR-containing 22Rv1 cells compared to the AR-devoid PC3 in vitro could be a result of miR-135b regulation of AR [25,26]. 

The top identified anti-miRNA in lung metastases and the only anti-miRNA enriched at all metastatic sites as well as at the primary site, blocks miR-23b, a known PCa metastasis suppressor [27]. A global miRNA reduction has been reported in PCa progression [28], and many are considered tumour suppressive, which is why we selected this screening method focusing on the tumour-suppressive miRNAs. This is why the oncogenic miRNA that was reduced was not studied further here, but it is noteworthy that several oncogenic miRNAs that have previously been described to be enriched in PCa bone metastasis were reduced in the bone metastases such as miR-410, miR-195, miR-127, and miR-335 [29,30,31,32]. 

Most metastasizing cells that enter the bone marrow are transient; only a few may remain for many years in a dormant state, and even fewer are activated and establish metastases. The reciprocal interactions between PCa and bone cells support the establishment and orchestrate the expansion of malignant lesions in bone. Advances in understanding the bone cell–tumour cell interactions can lead to novel means of controlling metastasis to bone. We found the PCa cells with blocked miR-379 to be strongly enriched in the bone of the mice and hence focused our attention to this miRNA, which has not been shown to inhibit PCa metastases previously. miR-379 has been suggested to act as a tumour suppressor in other cancer types; e.g., it has been reported that miR-379 inhibits migration and invasion in osteosarcoma and NSCLC targeting EIF4G2 [33,34], and inhibits tumour invasion and metastasis by targeting FAK/AKT signalling in hepatocellular carcinoma and gastric cancer [35,36]. We have previously shown that prostate cancer patients with low levels of unedited miR-379 had significantly shorter overall survival [37]. Our results show the effect of anti-miR-379 on cell growth in line with the enrichment seen in the primary tumours and in the paired patient data set. Further, there was a strong effect on establishment of metastases in the bone environment and colony formation in bone-like conditions. Measuring the miR-379 levels in patient samples, we found miR-379 to be lower in bone metastases than in the primary tumours. Our results also highlight the concept that, for miRNAs, their functional role is not only defined by their sequence but is also highly dependent on the cellular context. Considering the big pool of potentially actionable targets, the actual function of miRNAs depends on the target pool present in the cell itself. The abundance of the targets affects the probability of miRNA binding, this could be one of the reasons that the mRNA targets identified in the Ago2-IP differ in the two different settings. Thus, a specific miRNA can have different effects in different tissues as seen here, where miR-379 has different effects on the colony formation in different growth mediums. The in vivo validation gave clear evidence of introduction of anti-miR-379 in 22Rv1, which does not spontaneously metastasize to bone, leading to development of bone-metastases.

## 5. Conclusions

Our data suggest that specific miRNAs can drive the establishment of primary tumours and the dissemination of PCa cells to different distant metastatic sites. We have here identified a possible novel mechanism, where reduced miR-379 levels promote the progression of prostate cancer bone metastases. 

## Figures and Tables

**Figure 1 cancers-15-03892-f001:**
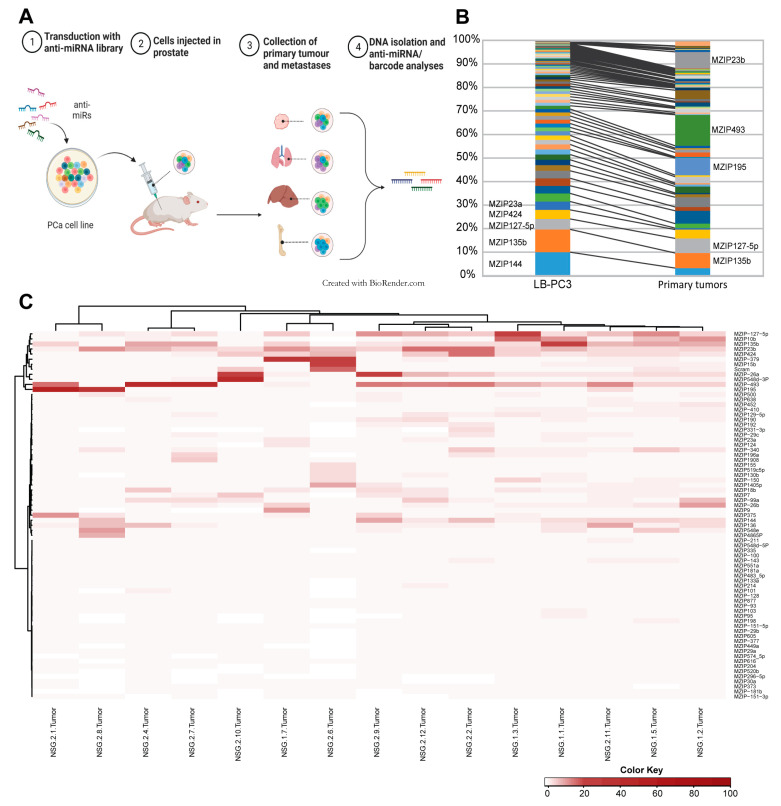
(**A**) In vivo functional screening. Illustration of the principle of enrichment of anti-miRNAs driving the PCa dissemination in the mouse model. Created with BioRender.com. (**B**) Average frequency of cells containing specific anti-miRNA inserts in primary tumours in relation to the injected cell pool. The most frequent anti-miRNA inserts are labelled next to the relevant bar. (**C**) Clustering analysis of the anti-miRNA content of primary tumours collected from the orthotopic mouse model. The analysis was carried out using Ward’s minimum variance method, with the cut-off set to 1%. The frequency of anti-miRNA inserts in individual primary tumours shows a common pattern among the samples.

**Figure 2 cancers-15-03892-f002:**
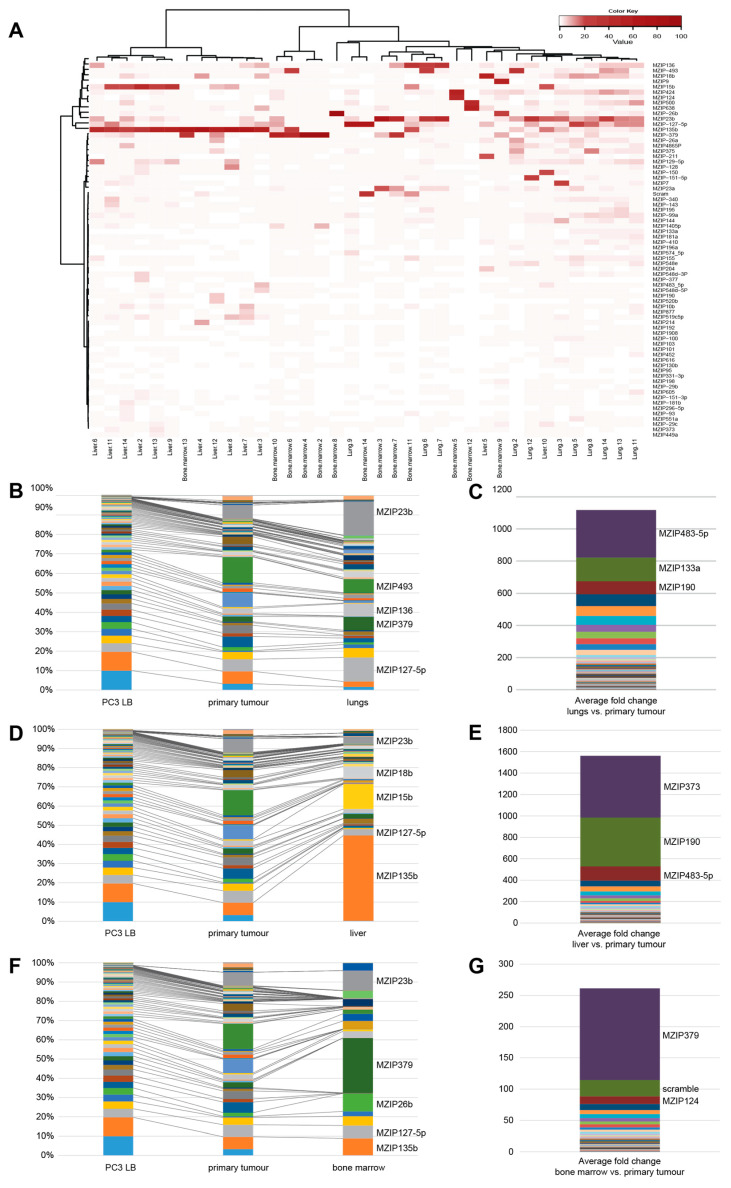
(**A**) Enrichment of individual anti-miRs at different metastatic sites. Clustering analysis of samples of metastatic sites collected from orthotopic mouse model including livers, lungs, and bone marrow. The analysis was carried out using Ward’s minimum variance method, with the cut-off set to 1%. (**B**) Average frequency of cells containing specific anti-miRNA inserts in the injected LB-PC3 pool, primary tumours, and lungs. The most frequent anti-miRNA inserts are labelled next to the relevant bar. (**C**) Average fold enrichment of specific anti-miRNA inserts in the lungs compared to their frequency in the corresponding primary tumour of individual mice. (**D**) Average frequency of cells containing specific anti-miRNA inserts in the injected LB-PC3 pool, primary tumours, and livers. The most frequent anti-miRNA inserts are labelled next to the relevant bar. (**E**) Average fold enrichment of specific anti-miRNA inserts in the livers compared to their frequency in the corresponding primary tumour of individual mice. (**F**) Average frequency of cells containing specific anti-miR inserts in the injected LB-PC3 pool, primary tumours, and bone marrows. The most frequent anti-miRNA inserts are labelled next to the relevant bar. (**G**) Average-fold enrichment of specific anti-miRNA inserts in the bone marrows compared to their frequency in the corresponding primary tumour of individual mice.

**Figure 3 cancers-15-03892-f003:**
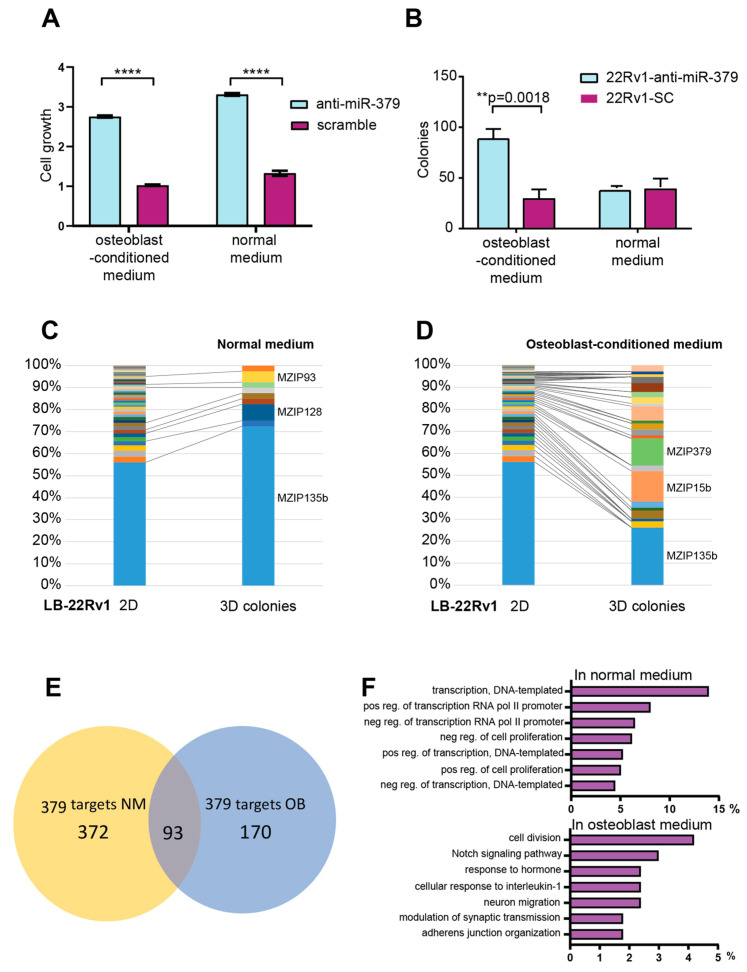
(**A**) In vitro functional analyses. Increased cell growth in 22Rv1 cells with anti-miR-379 compared to scramble control as determined with SRB assay. Performed in osteoblast-conditioned media or normal media. The results were confirmed in three independent experiments. **** *p* < 0.0001 (**B**) Increased colony formation in a colony-forming assay for 22Rv1-anti-miR-379 compared to 22Rv1-SC, in OB-CM, but not in normal growth media. The results were confirmed in three independent experiments. One replicate is shown here, and another is shown in Appendix A. ** *p* < 0.01 (**C**) Distribution of miRZip library inserts in analysed colonies from cultivation on agar plates (denoted 2D) or soft agar colony forming assay (denoted 3D) with LB-22Rv1 cell line in regular growth medium. Results are presented as average proportion of specific miRZiP insert-containing colonies from two independent experiments. (**D**) Distribution of miRZip library inserts in analysed colonies from cultivation on agar plates (denoted 2D) or soft agar colony-forming assay (denoted 3D) with LB-22Rv1 cell line in OB-CM. Results are presented as average proportion of specific miRZiP insert-containing colonies from two independent experiments. (**E**) Venn diagram of the overlap between miR-379 targets determined by Ago2-RIP-Seq, 22Rv1-anti-miR-379 compared to 22Rv1-SC in normal growth medium (NM) and osteoblast-conditioned medium (OB). (**F**) Top pathways identified by DAVID (http://david.abcc.ncifcrf.gov) on the differentially expressed genes being decreased upon anti-miR-379 expression in normal medium or osteoblast-conditioned medium.

**Figure 4 cancers-15-03892-f004:**
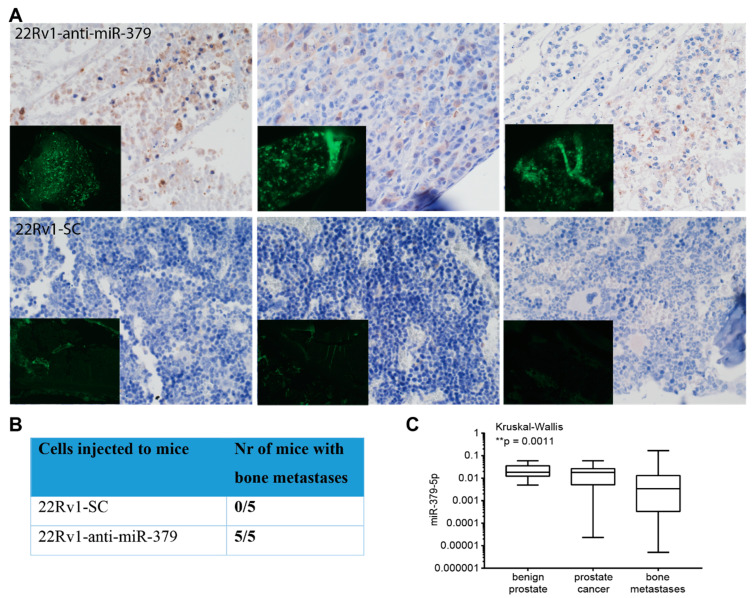
(**A**) In vivo functional analyses. Immunohistochemical staining with anti-turbo-GFP in femur of three representative mice injected intracardially with 22Rv1-anti-miR-379 (**upper row**) and 22Rv1-SC (**lower row**) viewed in 20× magnification. GFP-positive PCa cells are visible in the bone of anti-miR-379 mice but not the control mice. The inset picture is GFP fluorescence (green) in the adjacent section viewed in 10× magnification. (**B**) Table indicating the results of in vivo metastasis model in terms of bone metastasis development including 10 mice. (**C**) miR-379 expression levels in the patient cohort collected at Umeå University Hospital, constituting 55 bone metastases, 12 primary prostate tumours, and 13 non-malignant adjacent prostatic tissues.

## Data Availability

The data presented in this study are available in this article and Appendix A.

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
