# Peer review of "Functional In Vivo Screening Identifies microRNAs Regulating Metastatic Dissemination of Prostate Cancer Cells to Bone Marrow"

_cancers, 2023, doi:10.3390/cancers15153892_

Round 1

Reviewer 1 Report

The authors have used an anti-miRNA library in a functional in vivo screening assay to identify miRNA´s involved in the prostate cancer development and metastatic spread. The study involves both cell lines and in vivo experiments in mice, and the results found are correlated to findings in patient materials in a nice way. I find the study very well performed and presented in a nice way. I only have two minor questions:

1.      In Figure 4C and D the columns are labelled 2D and 3D colonies. I cannot find any explanation of how the 2D and 3D colonies are obtained neither in the Materials and Methods nor elsewhere, so it would be nice if the authors could explain this.

2.      In the first section of the Discussion the authors state that “The level of blocking of the anti-miRs cannot be evaluated directly due to its transient nature…”. Could the authors please elaborate a bit on what they mean by this, as the anti-miR procedure with transduction followed by antibiotic selection is usually considered to produce stable cell lines and not transient. I think the authors aim at that the inhibition of the miRs are transient in its nature, but I find the word “transient” a bit confusing as presented now in the context.

Author Response

The authors have used an anti-miRNA library in a functional in vivo screening assay to identify miRNA´s involved in the prostate cancer development and metastatic spread. The study involves both cell lines and in vivo experiments in mice, and the results found are correlated to findings in patient materials in a nice way. I find the study very well performed and presented in a nice way. I only have two minor questions:

  1. In Figure 4C and D the columns are labelled 2D and 3D colonies. I cannot find any explanation of how the 2D and 3D colonies are obtained neither in the Materials and Methods nor elsewhere, so it would be nice if the authors could explain this.

-Thank you for noticing. 2D is simply on a standard agar plate, while 3D is in a soft agar setting. We have now clarified this in the figure legends.  

  1. In the first section of the Discussion the authors state that “The level of blocking of the anti-miRs cannot be evaluated directly due to its transient nature…”. Could the authors please elaborate a bit on what they mean by this, as the anti-miR procedure with transduction followed by antibiotic selection is usually considered to produce stable cell lines and not transient. I think the authors aim at that the inhibition of the miRs are transient in its nature, but I find the word “transient” a bit confusing as presented now in the context.

      -The point we want to make is as the reviewer guessed the nature of the anti-miR inhibition. It binds the miRNA and hence blocking the function, but during a PCR the heating will cause the two RNA strands to separate and hence it is not possible to measure directly. We have rewritten the paragraph to clarify what we mean.

Reviewer 2 Report

In this manuscript Tina et al. performed in vivo functional screen to o identify microRNAs (miRNAs) regulating metastatic dissemination of PCa cells. They showed several miRNAs may play key roles in the metastasis of PCa cells. They also conducted a series of in vivo and in vitro studies on miR-379, and found miR-379 inhibits colony formation of PCa cell line in condition mimicking bone microenvironment. Overall, this study provides new insights on prostate cancer cells bone marrow metastasis. However, there are some issues that need to be discussed further. Therefore, I would like to recommend acceptance of this manuscript after major revisions. 

1) The mechanism of miR-379 in regulating PCa cells colony formation and metastasis is still unclear.

2) Some of data in figure 3 and figure 4 are of poor quality. For example, In Figure 3B, the authors should show the clone formation pictures.

3) In general, this article showed few experimental pictures, but it showed many summary pictures. the authors should added the experimental pictures.

4) There are two figure 4, but lack figure 3; and missing FIgure 3F.

5) In figure 4C, is there a difference of  miR-379 expression between bone metastases group and primary prostate tumours group? If there is no difference, how can the explanation be made? 

6) The materials and methods are written in a simple manner, for example, information such as primers for qPCR is lacking.

Author Response

In this manuscript Tina et al. performed in vivo functional screen to o identify microRNAs (miRNAs) regulating metastatic dissemination of PCa cells. They showed several miRNAs may play key roles in the metastasis of PCa cells. They also conducted a series of in vivo and in vitro studies on miR-379, and found miR-379 inhibits colony formation of PCa cell line in condition mimicking bone microenvironment. Overall, this study provides new insights on prostate cancer cells bone marrow metastasis. However, there are some issues that need to be discussed further. Therefore, I would like to recommend acceptance of this manuscript after major revisions. 

1) The mechanism of miR-379 in regulating PCa cells colony formation and metastasis is still unclear.

Yes, that is correct. We are reporting the finding that it has an effect on metastatic dissemination to bone, and then the indications which processes it might be effecting that can give that phenotype. The exact molecular mechanisms are bound to be a multitude of finely balanced events that we do not know.

2) Some of data in figure 3 and figure 4 are of poor quality. For example, In Figure 3B, the authors should show the clone formation pictures.

- We have improved the quality of the photographs in Fig 4. All figures can also be submitted separately or in other formats for higher resolution. We have included a picture from one of the repeats of the colony formation as supplementary figure 6.

3) In general, this article showed few experimental pictures, but it showed many summary pictures. the authors should added the experimental pictures.

Unfortunately, most of the experiments performed does not make illustrative pictures. We will take this to heart for future publications. We have added an experimental picture of the soft agar colony formation in Suppl figure 6.

4) There are two figure 4, but lack figure 3; and missing FIgure 3F.

This has been corrected!

5) In figure 4C, is there a difference of miR-379 expression between bone metastases group and primary prostate tumours group? If there is no difference, how can the explanation be made? 

I am mortified to find that it is the wrong graph in the manuscript we submitted! The correct graph has now been uploaded, that is corresponding to the text. Here there is a significant trend (as evaluated by Kruska-Wallis rank test) for decrease from benign, to primary to bone metastases). FYI, the incorrect graph was the expression levels of miR-379 in the Taylor dataset and contains a mix of bone and other metastases, which could explain why the difference is less pronounced, and not relevant for illustration for this publication.  

6) The materials and methods are written in a simple manner, for example, information such as primers for qPCR is lacking.

- The sequence of the PCR primers and sequencing primers are provided in Suppl Table 1. The qPCR primers are commercial, and the manufacturer and references are stated. The reference number for the miR-379 has been added.

Reviewer 3 Report

In this study, the authors identified a group of miRNAs involved in the establishment of primary tumors and the dissemination of prostate cancer (PCa) cells to different metastatic sites. To study the effect of miRNAs, they knocked down the miRNAs in the cells using anti-miRNA Lentivirus Library, also known as miRZIP, in PC3 cell line. The transfected cells were injected in the mouse prostate. This in vivo functional screening approach allowed them to identify individual miRNAs potentially inhibiting the metastatic process.

In particular, they focused on miR-379, which was found significantly lower in bone marrow metastases than to the primary tumor, suggesting its role as tumor suppressor, just as reported in the literature for other type of tumors. To elucidate its involvement in tumor progression, they performed an in vitro validation where the reduced expression of miR-379 was able to reduce the tumor growth in 22Rv1 cells cultured in osteoblast-conditioned medium (OB-CM) and normal serum, and the colony formation only under OB-CM condition. Furthermore, the better understand the molecular mechanism of action of miR-379, they performed Ago immunoprecipitation in 22Rv1 cells cultured in normal and OB-CM serum, followed by RNA-Seq. The analysis showed that 93 mRNA were found target of miR-379 in both conditions, and the gene ontology analysis of deregulated mRNA upon anti-miR-379 expression showed an enrichment of regulation of transcription and proliferation pathways in normal serum, while cell division, Notch signaling pathways in OB-CM condition.

Finally, they confirmed its activity by in vivo validation, where 22Rv1 cell line, known to their lower ability to develop metastasis, was transfected with anti-mir-379 and scramble followed by injected in NOG mice. Mice injected with anti-mir-379 cells developed bone metastasis compared to the same cell line transfected with scramble. 

The proposed mechanism is interesting and well supported by data. However, few experiments and clarifications would strengthen this manuscript.

Comments:

1) In this study, the authors used RIP-Seq approach to identify putative target of miR-379. They analyzed the differentially expressed target identified in normal serum and OB-CM. Why were the mRNAs common to both conditions not considered for pathway analysis? Is it possible to carry out a gene ontology analysis also on that group?

2) In the analysis of miR-379 expression levels in the patient cohort, the authors showed a significant reduction of selected miRNA in primary tumor and bone metastases compared to normal prostate, but there is no significant reduction in both tumor groups, unlike what was observed in in vivo functional screening. How the authors could explain this? Is it possible that miR-379 may play an essential role only in tumorigenesis? To further validate the activity of the selected miRNA, I would suggest performing RNA-Seq in patient samples as well and comparing the expression of miRNA-379 target with those identified by AGO-RIP-Seq. 

3) Since the authors did not observe any effect of anti-miR-379 on cell migration toward osteoblasts and considering what they observed from the expression level analysis in the clinical cohort, it is likely that the miRNA itself does not lead the increase of tumor aggressiveness. How the authors could explain this point? Is it possible that the activity of miR-376 requires the action of multiple miRNA.

4) An overall survival analysis should be performed on the same patient groups or by investigating TCGA database.

Minor Comments:

1) The authors indicated in the text that they used RIP-Seq approach, but in material and method, specifically in DNA sequencing section, they indicated to have used the library kit for chromatin immunoprecipitation (ChiP-Seq). Can the authors clarify this point?

2) Page 11 – It is Figure 3 instead of Figure 4

Author Response

Comments:

1) In this study, the authors used RIP-Seq approach to identify putative target of miR-379. They analyzed the differentially expressed target identified in normal serum and OB-CM. Why were the mRNAs common to both conditions not considered for pathway analysis? Is it possible to carry out a gene ontology analysis also on that group?1

The focus was on why miR-379 effected metastases to the bone and not other locations. The overlapping genes are not excluded from the analyses. But a separate analyses for only the genes targeted on both condition has been performed. They were found to be enriched for signal transduction, positive regulation of transcription from RNA polymerase II promoter and negative regulation of cell proliferation.  This information has been added to the section.

2) In the analysis of miR-379 expression levels in the patient cohort, the authors showed a significant reduction of selected miRNA in primary tumor and bone metastases compared to normal prostate, but there is no significant reduction in both tumor groups, unlike what was observed in in vivo functional screening. How the authors could explain this? Is it possible that miR-379 may play an essential role only in tumorigenesis? To further validate the activity of the selected miRNA, I would suggest performing RNA-Seq in patient samples as well and comparing the expression of miRNA-379 target with those identified by AGO-RIP-Seq. 

Our functional analyses do suggest that miR-379 can affect both the tumour initiation (cell growth) and the establishment at secondary sights (colony formation). This is supported by the patient data. However, I am mortified to find that it is the wrong graph in the manuscript we submitted! The correct graph has now been uploaded, that is corresponding to the text. Here there is a significant trend (as evaluated by Kruska-Wallis rank test) for decrease from benign, to primary to bone metastases). FYI, the incorrect graph was the expression levels of miR-379 in the Taylor dataset and contains a mix of bone and other metastases, which could explain why the difference is less pronounced, and not relevant for illustration for this publication.  

It would have been great to have access to fresh patient material from both primary tumours, bone metastases and metastasis in e.g. liver to make a RNA-Seq on patients. But unfortunately, we do not. Personally, I think that too much focus is put on identifying individual targets of miRNAs. They seem to work by regulating a multitude of targets to a low degree. And this seem to vary depending on the microenvironment. This is why we try to focus more on the effect on cellular phenotypes and pathways than individual targets.

3) Since the authors did not observe any effect of anti-miR-379 on cell migration toward osteoblasts and considering what they observed from the expression level analysis in the clinical cohort, it is likely that the miRNA itself does not lead the increase of tumor aggressiveness. How the authors could explain this point? Is it possible that the activity of miR-376 requires the action of multiple miRNA.

Again, sorry for the mixup of the patient graph. It is our belief that miR-379 in collaboration with other miRNAs contributes to the initiation of tumorigenesis by a modest effect on cell growth. In addition, we think it contributes to establishing bone metastases not by having an effect on migration but by effecting the colony-formation capacity in this setting.

4) An overall survival analysis should be performed on the same patient groups or by investigating TCGA database.

Unfortunately we do not have access to enough survival data from this cohort to be able to make a meaningful analyses. The TCGA cohort only contains 1 metastatic prostate cancer sample making this analysis less meaningful. However, we have previously performed and published a survival analyses in another cohort and found that low levels of unedited miR-379 had significantly shorter overall survival (Voss G et al. 2021; https://pubmed.ncbi.nlm.nih.gov/34433636/). We have now added this information and reference to the discussion.

Minor Comments:

1) The authors indicated in the text that they used RIP-Seq approach, but in material and method, specifically in DNA sequencing section, they indicated to have used the library kit for chromatin immunoprecipitation (ChiP-Seq). Can the authors clarify this point?

Thank you for noticing this discrepancy, it has been clarified that is RIP-Seq that has been performed.  

2) Page 11 – It is Figure 3 instead of Figure 4

Thank you for noticing. This typo has been corrected.

Round 2

Reviewer 2 Report

Despite some revisions by the authors, some problems remain. 

1) Although the authors provide a picture from one of the repeats of the colony formation as supplementary figure 6. However, the number of clones in the picture (supplementary figure 6) does not match the statistics (Figure 3B). 

2) Why most of the experiments performed does not make illustrative pictures. The authors should provide a picutre corresponding to Figure 4B. 

Author Response

Reviewer 2:

Despite some revisions by the authors, some problems remain. 

1) Although the authors provide a picture from one of the repeats of the colony formation as supplementary figure 6. However, the number of clones in the picture (supplementary figure 6) does not match the statistics (Figure 3B). 

- This is true! The Anchorage-independent colony formation in soft agar was repeated three times giving similar results, the graph depicts one of the replicates and the photo another (I have clarified that now!). In the graph the p-value is 0,018 in OBCM (no significance in normal medium) and in the photo the p-value is 0,017 in OBCM and no significance in normal medium. Hence showing very similar results.

2) Why most of the experiments performed does not make illustrative pictures. The authors should provide a picutre corresponding to Figure 4B. 

- Figure 4B is a summary of the number of mice with metastases. The presence of metastases was detected by GFP fluorescence and immunohistochemistry. This is what is shown in Figure 4A (for three of the mice in each group). It would of course have been more illustrative to perform some kind of whole body scanning e.g. using IVIS, but unfortunately, we did not have access to this equipment at that time.